# Innovation Systems and Sustainability. Development of a Methodology on Innovation Systems for the Measurement of Sustainability Indicators in Regions Based on a Colombian Case Study

Jhon Wilder Zartha Sossa [1,*], Juan Fernando Gaviria Suárez [2], Natalia María López Suárez [2], José Luis Solleiro Rebolledo [3], Gina Lía Orozco Mendoza [1] and Valentina Vélez Suárez [2]

1   Agro-Industrial Engineering Faculty, School of Engineering, Pontifical Bolivarian University, Medellin 050031, Colombia
2   Master of Sustainability, School of Engineering, Pontifical Bolivarian University, Medellin 050031, Colombia
3   Institute of Applied Sciences and Technology, National Autonomous University of Mexico, Mexico City 04510, Mexico
*   Correspondence: jhon.zartha@upb.edu.co; Tel.: +57-312-7807361

**Abstract:** In recent years, the need for changes in the current consumption and development patterns has become evident. To achieve this, it is necessary to innovate, create, and devise new methodologies and ways of thinking that allow for a reorientation towards economically prosperous, socially equitable, and environmentally reasonable practices. The purpose of this study is to identify sustainability variables within the framework of the innovation system concept, and to propose a methodology for diagnosing regions and identifying their gaps in a sustainability-oriented innovation system. The methodology was based on a literature review of different documents, where sustainability variables related to innovation were extracted from this search, consisting of the identification of concepts made by different authors regarding what should be considered an innovation system aligned with sustainable development. These concepts were identified as study variables and a questionnaire was formulated based on them, which was reviewed by experts to determine their relevance and congruence. After obtaining the final questionnaire, which was subsequently referenced as the diagnostic tool, it was applied in a Colombian study region consisting of the states of Quindío and Risaralda, with sustainability stakeholders. As a result, a great lack of knowledge on the subject on the part of the respondents, low accessibility to sources of information, and a high percentage of disarticulation between policies were found, which led to the conclusion that the evaluated region has a low understanding of sustainability. Based on the information discussed in this research, it can be concluded that there are currently no specific methodologies to measure the sustainability of a region or territory. Therefore, the tool is determined to be a guide for the measurement of sustainability in the context of innovation in any region. Finally, from the studies reviewed, the potential to include sustainability in the innovation systems of a region was detected, enabling economic development, the production of goods and services, and strengthening the socio-environmental considerations involved in the adequate use of natural resources and the increase in the quality of life.

**Keywords:** innovation system; sustainable development; sustainability; sustainable policy; literature review

## 1. Introduction

Sustainability aims to obtain equal and integral development among social, environmental, and economic determinants in a society; a goal that, in practice, may become a utopia [1]. In 2015, after the signing of 193 heads of state and governments around the world, the United Nations approved the 2030 Agenda and the Sustainable Development Goals (SDGs) to reduce the gap between aspirations and reality, making them a globally

accepted path to achieve sustainability [2]. The agenda recognizes that Science, Technology, and Innovation (STI) are key drivers enabling and accelerating global transformation toward prosperous, inclusive, and environmentally sustainable economies in developing and developed countries [3].

Additionally, sustainability has become a significant competitive differentiator for organizations, adding social and environmental value through the implementation of efficient technologies as a form of innovation [4]. Many innovations address the economic, social, and environmental dimensions of sustainable development. Among the types of innovation that can be found are materials and products that are inexpensive, durable, repairable, reusable, recyclable, and biodegradable, with greater affordability and lower environmental impact [5].

Integrating sustainability as part of a company's corporate, competitive, and innovation strategy is fundamental to its success [6]. Therefore, scientific and technological research allows for the identification of several application segments of sustainability innovation systems, where industry in general has the greatest involvement [7].

This research on innovation sought to assess the sustainability of a particular region (Risaralda and Quindío in Colombia) revealing the historical importance of the so-called "Coffee Triangle" (Caldas, Risaralda and Quindío) in the Colombian national development; the following study will focus only on the states of Risaralda and Quindío, assessing their competitiveness from the perspective of technology, science, and innovation as promoters of sustainable development. The performance and development of this study seeks to objectively assess the sustainability of this particular region. The methodology was based on eight steps, starting with the identification of stakeholders and ending with the analysis of the results of the tool's application, with particular emphasis on the literature review of scientific, technical, and legal support that supports the trustworthiness and coherence of the tool. As a result, sustainability variables were obtained, which consisted of identifying the affirmations or definitions stated by different authors regarding what should be considered in a sustainable development model. After identifying the variables, the construction and development of qualitative and quantitative questions were carried out, both of multiple choice, to facilitate their evaluation by interested parties, which were reviewed by experts in the sustainability field, applying the Delphi methodology by assessing its relevance and congruence [8–11]. Subsequently, the questionnaire was applied in the study region to evaluate its sustainability innovation and to propose this as a tool that could serve as a guideline for anyone interested in evaluating sustainability innovation in a region. This study was part of a master's thesis in sustainability and postdoctoral research related to innovation systems oriented towards sustainability.

Eventually, this research intends to be a methodology that will allow all interested in sustainability matters to find a way to assess it in an innovation context, and will contribute to solve the following question: how to establish and assess sustainability variables in regional innovation systems? Making it possible to perform evaluations with higher technical accuracy to make better decisions and improve the innovative capabilities of the system in question. The importance of developing this tool to evaluate sustainability resides in the opportunity it presents for small regions that do not have a large budget for assessment to have an approach to valuation with an easily replicable system based on current sustainability standards with innovation systems in mind.

## 2. Theoretical Frameworks

Innovation can be defined as the result of a set of activities that use knowledge, skills, and/or technologies to satisfy individual or social needs [12]. This requires efficient manufacturing processes aimed at increasing productivity, involving control technologies and improvement opportunities, providing benefits for human health and the environment, and leading to the efficient use of an organization's resources that will generate socioeconomic profits [5]. Innovation is the foundation of development, which requires restructuring institutional relations to consolidate innovation as a fundamental support for competitive and

sustainable advantages that will contribute to the ongoing creation of value [13]. A nation's competitiveness depends on its capacity to innovate and improve. Therefore, to preserve and expand national goods and services, innovation must be approached sustainably by developing tools to transition from disconnected and isolated processes to integrated and articulated processes [14].

According to Bermudez Estrada and Lara Coba [14] in Colombia, "sustainable innovation processes are sporadic, occasional, informal and unsystematic, adaptive and incremental innovations predominate, and Research and Development (R&D) activities are not frequent".

Two of the most relevant studies for this research were conducted by Zartha Sossa et al. [7], who aimed to collect and present the determinants of the Sustainable Innovation System (SIS) through qualitative data extracted through a systematic review of the literature using data from the Web of Science (WOS) and Scopus. On the other hand, the United Nations Conference on Trade and Development (UNCTAD) [5] proposed taking advantage of innovation for sustainable development through an in-depth review of Transformative Innovation Policies (TIP), in which a wide range of multidisciplinary actors are involved in promoting an inclusive and sustainable development agenda, thus promoting global development, which includes all sectors.

The following is a description of the fundamental concepts for the development of this study.

### 2.1. Sustainable Development-Sustainability

Sustainable development emerged in 1987 with the publication of the Brundtland report and involves meeting the needs of the present without compromising the ability of future generations [15] and ensuring a balance between economic growth, environmental care, and well-being [12].

Its definition warns of the negative environmental consequences of economic development and globalization and attempts to seek possible solutions to the problems arising from industrialization by involving significant long-term changes in lifestyles, technologies, infrastructure, and institutions [16]. Concepts such as eco-efficiency [17] and eco-innovation [18] in organizational culture guide all sectors of society to assume responsibility for sustainability and to apply knowledge and strategies to generate ecological improvements [19].

Resource mobilization and protection policies, which are necessary for the implementation of sustainability, are processes that involve continuous dialogue and participation to create alliances of trust [20], ensuring that the actions of companies and organizations are framed within the legal and juridical scope, and that they are adjusted and aligned with the different parameters of sustainability [21].

Consequently, the knowledge flow on how to develop sustainable systems demands conversion processes that must be available to anyone who requires them [21]. A vision of sustainability is developed in the new agenda for sustainable development, which seeks to end poverty, promote prosperity and well-being for all people, and protect the planet by 2030 [22].

### 2.2. Innovation System (IS)

At the internal level of each nation, there is the development of open, complex, and evolving systems, involving relationships within and between organizations, institutions, and socio-economic structures, which determine the pace and direction of skill development resulting from science- and experience-based learning processes [23,24].

To adapt to a constantly changing environment, many organizations in pursuit of meeting their objectives initiate exploration processes where experimentation and networking with people and entities within and outside them [25] constitute fundamental alliances and interconnections for the implementation of different forms of innovation [26]. Innovation is defined as the structuring of new ideas and knowledge that enables the improvement of products, processes, services, operations, and people in organizations [27]. Innovation involves discovering and detailing the use of strategies for system optimization [28]. Innovation requires a network of institutions whose activities involve the importation, modification, and diffusion of modern technologies [29]. Both innovation capacity and production competition influence research and development [30]. Consequently, Innovation and Development (I&D) is based on a strategic plan that involves the creation of current ideas, products, or services, leading to the development of areas such as education, technology, business models, and ecology [12].

Innovation systems have emerged as implementation tools in which the flow of information and technology between people, companies, and institutions is key to an innovative process [31,32]. Parallel to this is the interaction between the stakeholders necessary to convert an idea into a successful process, product, or service in the market [33]. The elements in an innovation system work as support for the promotion of learning processes related to the production of knowledge among individuals or agents of the system [34].

### 2.3. Sustainable Innovation Systems (SIS)

Some aspects of innovation can take the world in the wrong direction, directly opposite to a sustainable future, which is why ecological systems emerge as a fundamental connection between social systems and the surrounding environment [35]. In these systems, knowing, measuring, and generating connections with the diversity of their environment are fundamental to promoting and circulating the contents and making the information reach the entire interested sector [21]. Resilience is a precondition for sustainable development, referring to the ability of a system to withstand and overcome limiting situations [5] that can be affected and damaged by internal and external organizational disruptions and its stakeholders in its social and geographical environment [36].

Innovation in terms of sustainability linked to the 2030 agenda and its development goals is a guide for any stakeholder, called the government, business, or academic community, seeking to internalize and accommodate the economic, social, and environmental dimensions of sustainable development in their respective regions [37,38]. Achieving SDGs requires the commitment and collaboration of stakeholders with the capacity to design, develop, and implement innovations using bio-economic approaches [39]. The aforementioned stakeholders include businesses, entrepreneurs, educational and research actors, organizations that finance innovation, trade unions, and national and international donors such as NGOs [5].

SIS consists of natural elements, humans, and relationships that interact in the production, dissemination, and use of new knowledge that is economically useful [22]. In innovation management, various methodologies and techniques must be employed in the distinct phases that constitute the innovative process to cope with the associated risk and manage the progress appropriately to obtain sustainable efficiency in the processes and their relationship with stakeholders [40]. According to Zartha Sossa et al. [7], these methodologies and techniques are known as innovation management tools and seek to constitute methodologies that enable the correct development of systems such as the circular economy. This aims to reduce both the input of virgin materials and the production of waste, closing the "loops" or economic and ecological flows of resources [20], allowing companies to create a structure for sustainable innovation through environmental, economic, and social goals [41]. The connection between the actors involved in the innovative process inside and outside the company allows the development and application of different innovation

tools, creating synergy between the involved participants, who obtain more value in their products and processes than separately [21].

In this way, a SIS constitutes one of the main challenges for economic sectors, academia, and government: innovating sustainably and offering the possibility of increasing competitiveness in global markets while complying with international requirements and regulations [42].

## 3. Materials and Methods

This tool seeks to objectively assess the sustainability of a specific area in Colombia (Risaralda and Quindío). The methodology of the present study consists of the following steps:

### 3.1. Stakeholder Selection

- Public and private sectors (companies, associations, and private sector organizations).
- Educational community (human resources (teachers and administrative staff), educational organizations, academic communities, and research groups) [43].

### 3.2. Literature Review

- Articles were obtained from the SCOPUS database using the search equation, TITLE-ABS-KEY ("innovation system*") W/3 sustainab*.
- International agreements and consensus that set sustainability goals.
- National regulations related to sustainability [44].

### 3.3. Questionnaire Construction

From the literature review, sustainability variables related to or impacting innovation were extracted and then simplified. Based on the simplified sustainability variables, a series of questions were constructed with multiple answers to facilitate the responses of the interested parties while maintaining an intrinsic relationship with the established simplified variables, allowing for an objective and quantifiable assessment of sustainability in the target region.

### 3.4. Delphi Methodology (Evaluation of the Relevance and Congruence of the Questions in the Questionnaire)

To maintain a high level of competence in assessing the relevance of the questionnaire, experts were required to have a high level of knowledge and experience in the field of sustainability and innovation. The chosen ones had degrees such as master's in environmental sciences with emphasis on sustainability, master's in eco-audits and corporate environmental planning, and a doctor in technological innovation projects in engineering. All had a strong professional background and extensive research experience.

Congruence and relevance were evaluated as follows:

- Congruence: each expert assesses whether the content of each item included in the questionnaire reflects the specified objectives, "1" if the content is specified, "−1" if he/she believes it does not measure it, and "0" if there are doubts about whether it measures it [11].
- Relevance: each expert assesses whether the content of each item included in the questionnaire reflects the relevance of each item on a Likert scale from 1 to 5, to measure the proposed objective; from "not relevant at all" (1); to "totally relevant" (5) [11].

After the completion of the Delphi methodology and following the recommendations given by experts, the final questionnaire was finalized.

### 3.5. Development of the Questionnaire as an Objective Tool

Once the final questionnaire was obtained, it was shared with a group of stakeholders in the region (Risaralda and Quindío), which was carried out through an exhaustive search of people whose roles were related to sustainability:

- Academy
- Government
- Industry
- NGOs in the region

### 3.6. Application of the Tool (Questionnaire) in the Target Group

The tool was sent and distributed to the target population in the study region as an invitation to contribute to sustainability research through online platforms.

### 3.7. Analysis of the Application Tools Results and Coefficient of Expert Competence "K"

Once the survey application phase was completed, the information was analyzed using descriptive statistics to record conclusions and recommendations for future applications.

There has been great interest in analyzing and establishing the degree of expertise of the participants in in-depth surveys, especially in the Delphi method, a solution was presented by Cabero and Barroso [45] related to the calculation of the "coefficient of expert competence K" which "is made from the opinion shown by the expert on his level of knowledge about the research problem, as well as the sources that allow him to argue the established criterion" [45] cited by Zartha Sossa et al. [46], and Zartha Sossa et al. [10].

According to the authors, the coefficient of expert competence K is calculated using the following expression: $K = 1/2 (Kc + Ka)$.

Considering that:

- $Kc$= refers to the "coefficient of knowledge" or information that the expert has about the topic or problem posed. It is calculated from the valuation made by the expert himself on a scale of 0 to 10, multiplied by 0.1.
- $Ka$= related to the "coefficient of argumentation" or substantiation of the expert's criteria. This coefficient is obtained from the assignment of a series of scores to the various sources of argumentation that the expert has been able to wield [10,46].

According to the experience of the authors of this paper, it could be convenient to include new items or sources in $Kc$ and $Ka$, especially if the purpose is to obtain details about the years of experience of the participants and/or production or co-authorship of papers, patents, industrial secrets, and even participation in projects in the study area.

In the original methodology, based on the final values obtained, the experts are classified into three large groups:

- If K is greater than or equal to 0.8, then there is a high influence from all sources.
- If K is greater than or equal to 0.7, and less than 0.8, then there is a medium influence from all sources.
- If K is greater than or equal to 0.5, and less than to 0.7 then there is a low influence of all sources [45], cited by Zartha Sossa et al. [46], and Zartha Sossa et al. [10].

According to the guidelines of several authors cited by Cabero and Barroso [45], experts with values lower than 0.8 are not contemplated in the study and therefore are rejected, on this aspect in the experience of the authors, in previous studies when Delphi applications are performed in companies could include experts with a K value greater than or equal to 0.7 or use the range above 0.8 but considering new sources, contributions, co-authorships, and experiences of the participants. For this study, the survey results were compared between the unrestricted group and the group with a K value greater than 0.7.

Figure 1 shows a summary of the 7 steps performed for this study:

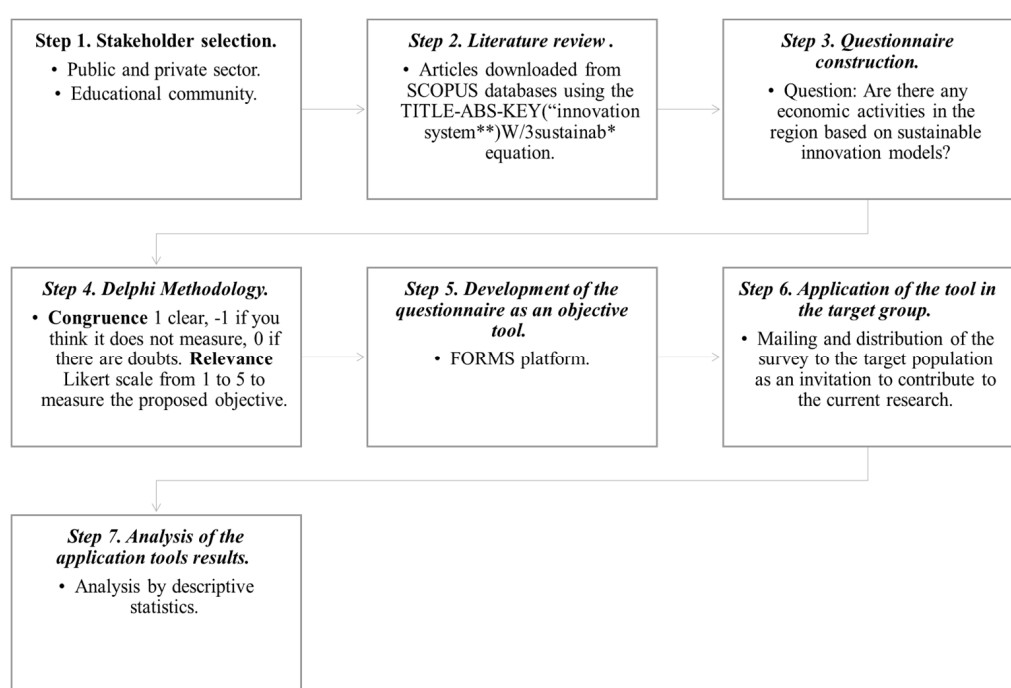

**Figure 1.** Descriptive image of the methodology (own elaboration, 2021).

## 4. Results

The literature review process yielded 92 study documents directly related to innovation and sustainability systems, and declarations from international summits such as the United Nations Agenda 2030 [47], in addition to public policies and national development strategies where the national circular economy [47] and the green growth policy [48] stand out. In Figure 2 the country of origin of the 92 study documents is shown. As can be seen, 15% of the documents correspond to proposals, projects, agreements, and studies formulated in Colombia, and their high consideration for the research is aligned with assessing the state of innovation in sustainability in the region studied.

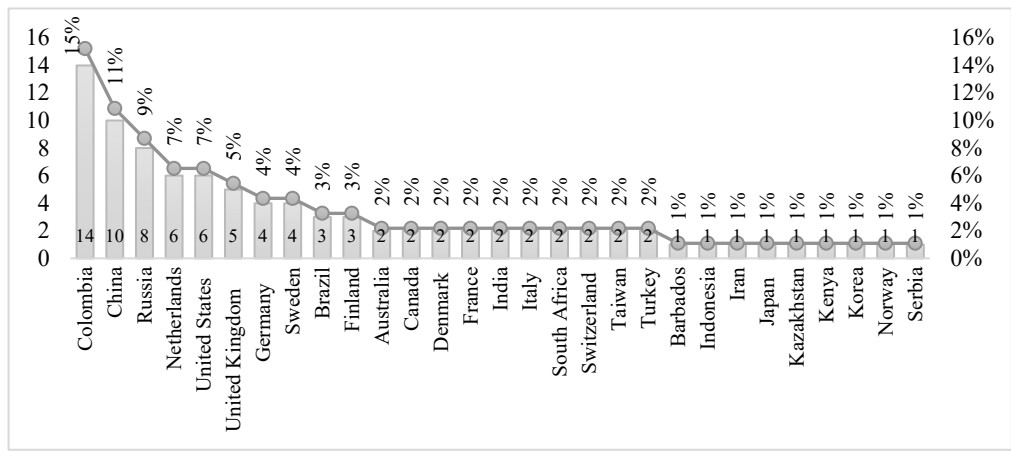

**Figure 2.** Provenance of literature review by country (own elaboration, 2021).

After reading the documents in depth, sustainability variables were identified that correspond to terminologies that define sustainable innovation and its implications for a total of 216 variables found, where they were also associated with a dimension of sustainability (social, environmental, and economic) that were simplified among similar ones to 51. Each of these 51 variables raised a question to determine the state of sustainability in the study region and their evaluation criteria were defined using a multiple-choice answer, this being the questionnaire tool.

Three experts on sustainability and innovation were selected to evaluate the congruence and relevance of each question. According to the consensus of the experts surveyed and applying the Delphi methodology, it was determined that 5 of the 51 questions initially proposed needed to be revised because they were not relevant or congruent with the research topic. The questions in the revision were adjusted according to expert recommendations and subsequently included in the questionnaire.

After the questionnaire was completed, the group of interest was selected, comprising 65 people, of whom 29 belonged to Quindío and 36 to Risaralda. The questionnaire was virtually applied. In Figure 3 the percentages of respondents by region are shown.

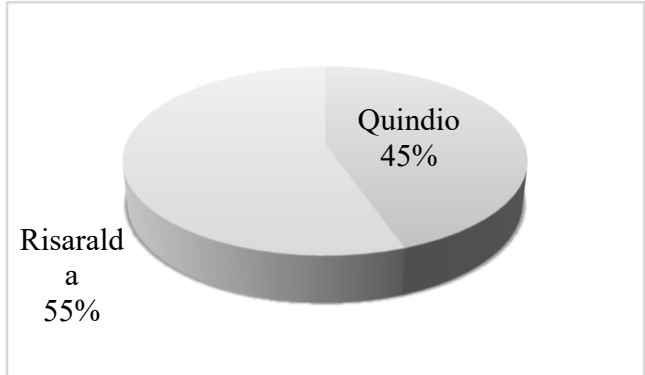

**Figure 3.** Percentage of respondents by region (own elaboration, 2021).

To determine the level of knowledge of the topic (Kc), respondents answered the following question: please select one of the items in which you consider that you are or are not knowledgeable about the topic (sustainability). Figure 4 shows the percentage of respondents according to their level of knowledge.

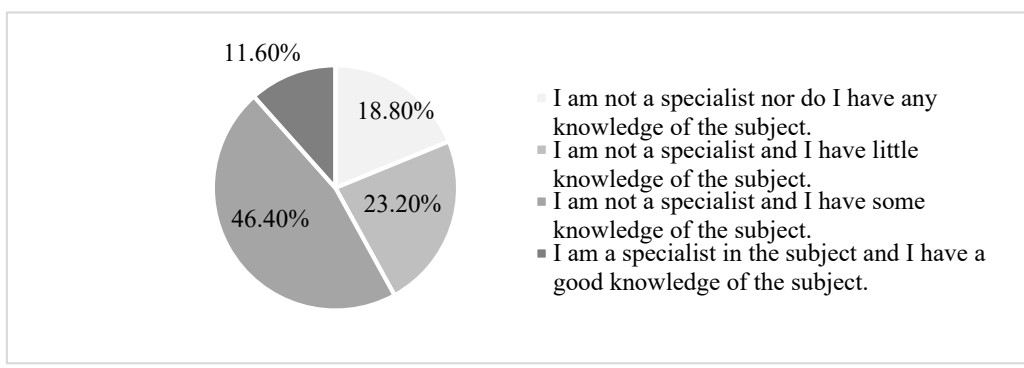

**Figure 4.** Source of knowledge Kc (own elaboration, 2021).

We found that 18.8% were neither specialists nor had any knowledge of the subject; 69.6% mentioned not being specialists but had little or some knowledge of the subject; 11.6% were specialists and had a fair amount of knowledge of the subject; and none of the respondents mentioned having total knowledge of the subject. The results obtained from the 51 questions asked to the 65 respondents were analyzed to obtain the percentage of consensus in each or of the departments of the selected region to be plotted according to Figure 5 and analyzed according to Table 1, and the individual responses of the respondents with greater and lesser knowledge of the topic in each area of the region.

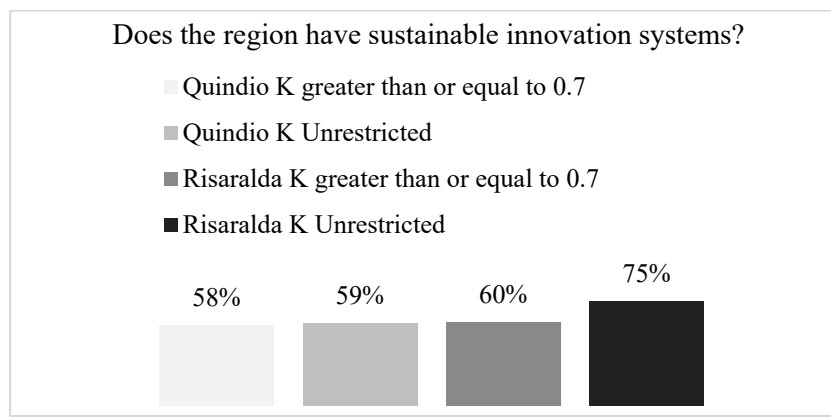

**Figure 5.** Example analysis graphic performed on the questions under consideration (own elaboration, 2021).

**Table 1.** Example of descriptive analysis performed on the questions under consideration (own elaboration, 2021).

| Ask | Does the Region Have Sustainable Innovation Systems? |
|---|---|
| Quindío K greater than or equal to 0.7 | More than 30% but less than 50% of their processes have sustainable innovation systems. |
| Consensus percentage | 58% |
| Quindío K without restriction | More than 30% but less than 50% of their processes have sustainable innovation systems. |
| Consensus percentage | 59% |
| Risaralda K greater than or equal to 0.7 | More than 30% but less than 50% of their processes have sustainable innovation systems. |
| Consensus percentage | 60% |
| Risaralda K unrestricted | More than 30% but less than 50% of their processes have sustainable innovation systems. |
| Consensus percentage | 75% |
| General K Greater than or equal to 0.7 | More than 30% but less than 50% of their processes have sustainable innovation systems. |
| Consensus percentage | 59% |
| General K Unrestricted | More than 30% but less than 50% of their processes have sustainable innovation systems. |
| Consensus percentage | 68% |
| Analysis | It is found that at a general level more than 30% but less than 50% of the processes in the region surveyed have sustainable innovation systems. |

Figure 6 shows that there was a representation of at least 20% of each of the three stakeholders (academy, industry, and state), with a greater representation of the academic sector of the two regions surveyed (Quindío and Risaralda), which associates the majority of responses obtained in this study from the perspective of the academy on sustainability and innovation in Colombia.

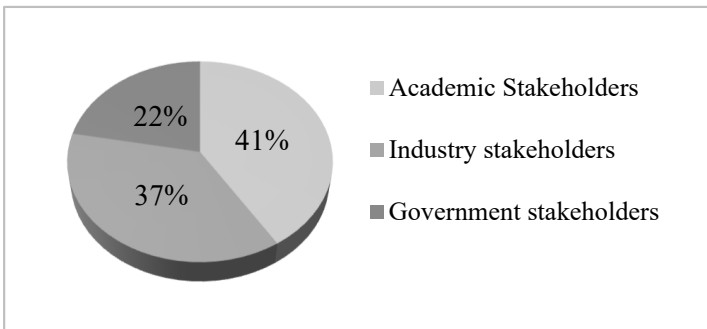

**Figure 6.** Distribution of stakeholders (own elaboration, 2021).

Figure 7 shows that only 4% of the surveyed industrialists represented by one person were considered specialists in innovation and sustainability issues, which could show the poor incorporation of these issues into the development of these issues in the region. In contrast, 46% did not consider themselves specialists and declared that they had little knowledge of the subject.

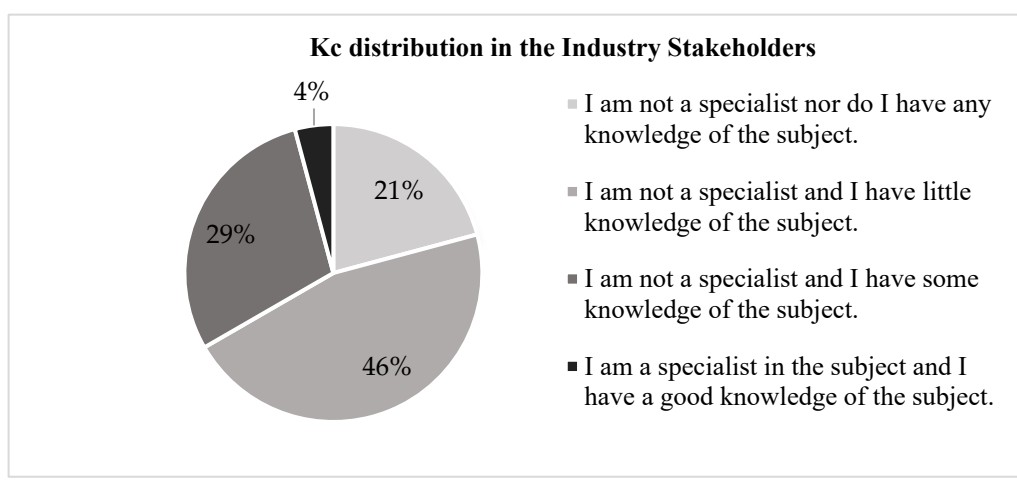

**Figure 7.** Kc distribution in industry (own elaboration, 2021).

Figure 8 shows that 19% of the academics surveyed, represented by five people, are considered specialists in the topics of innovation and sustainability, which shows that most of the specialists in the topic in question are dedicated to the academy and not to the field of physical application, as in the industry or government.

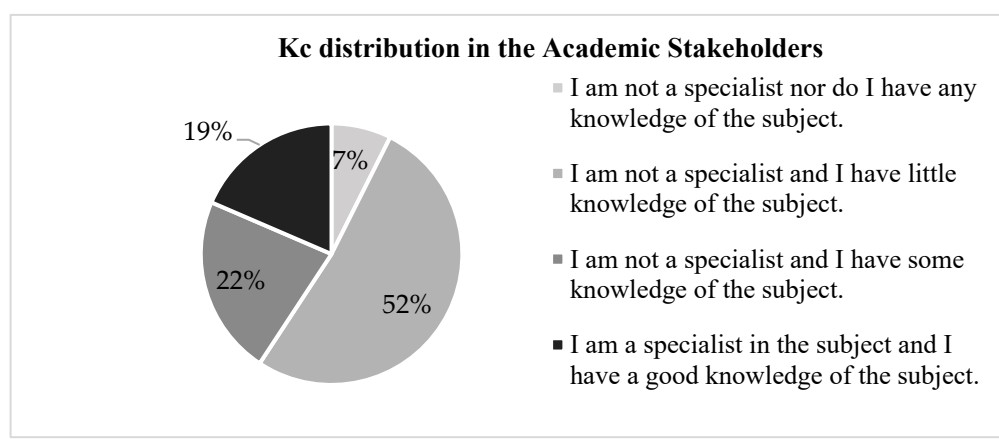

**Figure 8.** Kc distribution in academia (own elaboration, 2021).

In Figure 9, it is surprising that none of the people surveyed from the government considered themselves specialists in the issues of innovation and sustainability, which could indicate a significant failure in the governance of the territory in the face of the development of these issues. There is also a disarticulation between the government, academia, and industry on the subject because, according to the information presented in Figures 7 and 8, most experts in academia and knowledge do not reach the levels of execution with society through industry and government.

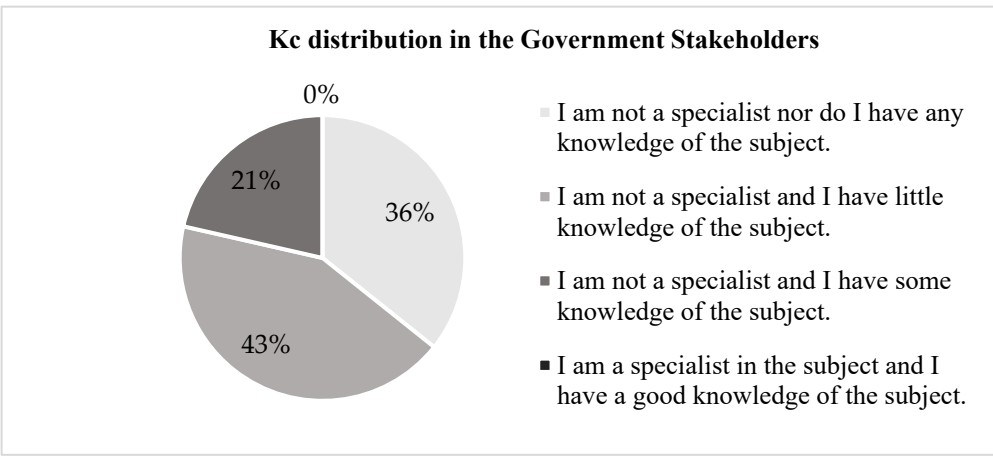

**Figure 9.** Kc distribution in the state (own elaboration, 2021).

At the generic level, the region has low percentages in the development of sustainability in the context of innovation, parallel to the low articulation evidenced between multiple sectors and their improvement systems.

## 5. Discussion

The discussion was conducted considering four axes: sustainability variables to be considered in innovation systems, limitations for the measurement of sustainability, innovation as a transversal axis to sustainability, and the last axis related to an application case for the measurement of a sustainable innovation system applied to the study region (Quindío and Risaralda) of Colombia.

### 5.1. Sustainability Variables to Be Considered in Innovation Systems

Sustainability has been the forgotten variable in the theory and practice of innovation systems, and according to the literature review conducted in this study on 92 papers, it was observed that 27 of them, although they analyzed the innovation and/or sustainability systems, did not delve into identifying or applying tools/methods/techniques to include sustainability in innovation systems. In the study by Manyweathers et al. [49], although they studied the sustainability of agricultural innovation systems, they did not identify specific sustainability variables that apply to sustainable innovation systems.

Wang et al. [50] used modeling of spatial econometrics, where they measured regional innovation associated with its performance and collaborative innovation in the research of the industry-university relationship. However, although he sustained the purpose of the study on the sustainability of regional innovation, he did not identify specific variables related to sustainability. The same can be observed in Makarova and Firsova [51], who discussed the functional interrelationships between the elements of a regional innovation system.

Purkus et al. [52] mentioned the variables identified by Hekkert et al. [53] and Bergek et al. [54] as entrepreneurial activities, knowledge development, knowledge diffusion through networks, orientation of search processes, market formation, resource mobilization, and legitimacy building. However, their study focused more on the technological innovation system of the German wood-based bioeconomy and its sustainabil-

ity. Laukkanen & Patala [55] analyzed barriers to the development of a sustainable business innovation model using the innovation system functions of Hekkert et al. [53] and Bergek et al. [54] to overcome regulatory, market, and behavioral barriers.

Kilkiş [56] identified innovation only concerning renewable energy, energy efficiency, and environmental management technologies in relation also to the variables identified by Hekkert et al. [53] and Bergek et al. [54] to represent the systemic interactions expected to take place within an SIS, similarly, could be observed in Kilkiş [57] where he measured and compared the SISs (Sustainable Innovation Systems) of four countries with emerging economies, where the focus of the research was the entrepreneurial activity for inclusion in innovation in clean technologies, but without taking into account different sustainability variables in the other aspects.

Van Someren and Van Someren-Wang [58] in their book "Strategic Innovation in Russia: Towards a National System of Sustainable and Rentable Innovation" mention sustainability within the arguments for adopting and applying the strategic innovation approach in Russia, but, although they identify a clear relationship between sustainability and innovation systems, no variables are found that would allow us to evaluate an SIS. The same could be observed in the studies of Rogers [59], Guo [60], Djeflat [61], Brent [62], Rubach [63], Li-li et al. [64], Lazzeretti et al. [65], Laster [66], Wit [67], Miller et al. [68], Bossink [69], Floricel et al. [70] and Gerstlberger [71], who measured innovation systems without considering sustainability variables or did not report explicit variables.

Now, 65 documents analyzed contributed sustainability variables and made explicit their relationship with innovation and/or sustainable innovation systems, which will be discussed further and synthesized.

Finally, Zartha Sossa et al. [7] raised the search for determinants necessary to transition from Innovation Systems (IS) to Sustainable Innovation Systems (SIS), where innovating sustainably offers the possibility of increasing the competitiveness of markets at a global level, complying with international requirements and regulations. However, as a complement to the study by Zartha Sossa et al., this study updates the concepts of sustainability and innovation through a literature review and develops a tool. Sustainability is measured in the context of innovation in a region through an analysis of multiple variables.

*5.2. Constraints to Sustainability Measurement*

Sustainability must be considered through the utility generated by new generations based on the understanding that natural capital is limited and cannot be replaced by human capital. This approach seeks to ensure a non-declining level of natural capital; that is, it implies that the stock of natural resources should remain constant over time with no substitutability between distinct types of capital [72].

Therefore, Lorenzo Báez [72] established that one of the main limitations for the measurement of sustainability is that the development of current strategies is based on the concept that environmental costs must be internalized and resources substituted by their monetary value, while total aggregate capital must be maintained or increased from one generation to the next, thus causing the so-called weak sustainability.

According to the study conducted in the region of Risaralda and Quindío, the weak sustainability proposed by Lorenzo Báez [72], is fueled by the lack of interconnection between processes, which directly prevents working on the development of systems in an articulated manner. In addition, very few processes have sustainable alternatives that promote the conservation and recovery of ecosystems and social goods and services, implying minimal progress in the region for the development of sustainability.

Consequently, strategies have been developed to bridge the gap between development and sustainability. Lee & Mwebaza [12] proposed sustainability measurement models as new tools capable of providing conclusive data when analyzing a landscape.

Parallel to this, Plasencia Soler et al. [73] established that a good model for measuring sustainability must have a multidimensional character in the concept of sustainability, thus

integrating social and environmental aspects as part of its components and decreasing the limiting factors in the measurement.

These measurement models are based on the principle that sustainability is a continuous and evolutionary process achieved by organizations through development in stages, from simpler to more complex levels. These models allow the construction of a hierarchy composed of diverse levels in which the processes of an organization evolve, which allows the inclusion of a fundamental dimension of the concept of sustainable development: the temporal dimension [73].

In contrast, Ermilova et al. [23] exposed a fundamental weakness of such models that they do not work under a unified measurement. Moreover, the effect of an operation has on society and the environment cannot always be expressed in numerical terms, making it difficult to measure them. Therefore, the aspects to be evaluated can affect the dimensions positively or negatively, and there is no universally defined measure of what is right or wrong or of the permissible range for many aspects that are evaluated: environmental, economic, and social [74].

Indicators for measuring sustainability represent a key factor to be analyzed, and Ibáñez Pérez [75] explains indicators as tools to clarify objective achievements and impacts, through which trends or certain phenomena are detected. They are regularly designed to have standards against which to evaluate, estimate, or demonstrate the progress of variables for established goals; Sustainability Indicators (SI) attempt to relate environmental information with economic and social information to generate information on pollution, deterioration of productivity development, or welfare achieved by the population [75]. Therefore, the poor management of such indicators constitutes a direct limitation to the measurement of sustainability at the regional level.

According to Firsova et al. [24], many of these indicators, after their application and execution, do not have a continuous follow-up of their results because most entities or regions do not have a structure for reception and analysis, resulting in null and untimely corrective actions to the results obtained, creating direct limitations to an adequate measurement of sustainability. The updated and easily accessible sources of information are the origins of the improvement of the measures implemented for the development of sustainability, from the application of the tool generated in the present study, it is found in the vast majority that the respondents are not specialists and have little or no knowledge of the subject.

For Ermilova et al. [23], open science provides greater transparency and reproducibility of results. Ecology and earth sciences likewise provide a baseline against which changes towards sustainability can be assessed. Open data and open science are sources of innovation, as they drive economic growth through the development of new systems and methodologies, which are the main objectives of governments and companies at distinct levels.

It is clear that multiple tools have been developed to perform an adequate measurement of sustainability, and with it transform activities that generate great changes in the conservation and maintenance of both social, economic, and environmental resources, however, UN and CEPAL [22] find as the main limitation, that such information has mostly limited access to the public, losing significant potential and is currently underutilized for reuse in new products and services; this prevents adequate information to citizens, improve governance and direct sustainable development.

### 5.3. Innovation as a Transverse Axis to Sustainability

The transformation of our world under the model of sustainability requires a vision of innovation that appreciates the contributions that can be achieved in the fulfillment of the 17 Sustainable Development Goals (SDGs). Science, technology, and innovation play an especially significant role in changing paths toward more inclusive and environmentally sustainable development patterns [5].

According to Abraham and Dao [26], innovation systems in sustainability refer to the investment and development of innovation systems to achieve sustainability, and innovation is not a variable of sustainability, but as a means to achieve it. The position shared by

Zartha Sossa et al. [7] states that to achieve the objectives of sustainable development, it is necessary to achieve innovation that allows the use of available technology to reduce the negative environmental impact resulting from productive activities, generating synergies between sectors, and thus achieving information and communication technologies that improve the performance of companies in the context of sustainability.

In contrast, according to Korhonen et al. [21], sustainable development, and as a reference, the 17 Sustainable Development Goals (SDGs) could direct and guide the establishment of innovation as a priority in regional policies, which does not establish innovation to achieve sustainability, but innovation as the desired consequence in the implementation of sustainable development.

It is not possible to define in this article whether innovation, science, technology, and sustainability are causes or consequences in a strict chronological order, but it is clear that their interrelations allow the synergies necessary for the achievement of sustainable development. Table 2 presents the authors who related innovation systems to sustainability, the variables detected in each of the texts, and the questions with their respective answers.

**Table 2.** Sustainable innovation systems evaluation tool (own elaboration, 2021).

| Simplified Variable | Reference | Questions |
|---|---|---|
| The proportion of the informal economy. | [5] | What percentage of the region's economy belongs to informality? |
| When talking about sustainable development, five dimensions must be considered (social, economic, environmental, spatial, and cultural) which must be supported by public-private partnerships for the development of the different sectors. | [5,16,21,32,76–79] | How many synergies exist between the dimensions of sustainability (social, environmental, economic, spatial, and cultural)? |
| Participation of a wide range of actors from all sectors is a fundamental process to building a sustainable innovation system and increasing the use of the internet and mobile networks provide an opportunity to improve access to knowledge that can accelerate the scaling-up process. | [5,47,80] | At the regional level, are there interconnections between the processes that allow working on the development of systems in an articulated manner? |
| Consolidate sustainable alternatives for production, conservation, and recovery of ecosystem goods and services. | [16,40,42,81,82] | Does the region have sustainable development alternatives that promote the conservation and recovery of ecosystem goods and services? |
| Adopt urgent measures to combat climate change and its effects. | [17,37,83–87] | Does the region have action measures that contribute to reducing climate change and contribute to the sustainability of the processes? |
| Protection and conservation of the set of environmental variables that constitute a good or service. | [16,37,88] | Does the region have regulatory norms for the care and conservation of environmental variables? |
| Continuous improvement is a fundamental variable for the development of new and better sustainable innovation systems. | [78] | Does the region have self-assessment processes in place to determine the status of the different sectors involved? |
| Research and development systems that allow process improvement and the creation of new patentable inventions. | [5,21,24,47,89–91] | Does the region have economic resources earmarked for the development of academic research? |
| Regional innovation system based on resource conservation, taking into account the form of resource integration, value orientation, and ecological considerations. | [92,93] | Does the region have sustainable innovation systems? |

**Table 2.** *Cont.*

| Simplified Variable | Reference | Questions |
|---|---|---|
| Absorption and adaptation capacity. | [31] | Does the region have technological systems that allow it to adapt to the different changes in the environment? |
| Urban space management, with adequate design and distribution of infrastructure. | [5,16,83,89,94] | Does the region have an adequate distribution of the POT for the development of its economic sectors and the location of its population? |
| Healthy life models. | [83] | Does the region develop plans and models that are disseminated for the knowledge and application of healthy living? |
| Develops programs to improve the working conditions of the organization's employees. | [95] | Do organizations in the region develop programs to improve the working conditions of the organization's employees based on sustainable models? |
| There is a correlation between the transition of a country's economic sectors towards sustainability and their competitiveness, framed in the regulatory and coordination processes adopted by the sectoral innovation system directed towards sustainability. | [42] | Are there economic clusters in the region based on sustainable innovation models? |
| Regional economic activities based on sustainable models. | [26,48,96–107] | Are there economic activities based on sustainable innovation models in the region? |
| Explore methodologies for recycling and utilization of carbon emissions, hand in hand with the implementation of low-carbon technologies. | [88,108] | Do technologies that reduce carbon emissions exist in the region? |
| Adaptation of policies to support sustainability. | [21,109] | Are there policies in place in the region to implement sustainable models in the communities? |
| Adoption of strategies for sustainable development based on poverty reduction and increased inclusion and equity. | [16,110] | Is there adoption of strategies for the development of sustainability based on poverty reduction and increased inclusion and equity in the region? |
| Promotes conflict resolution strategies and the search for peace and justice. | [84] | Are there strategies for conflict resolution and the search for peace and justice in the region? |
| Creation of internal support structures to promote sustainability. | [12,13] | Are there any public organizations in the region for the promotion of sustainability? |
| Active participation of stakeholders in the development of the organization's sustainability innovation. | [12] | Is there active stakeholder participation in the development of innovation-based sustainability of organizations in the region? |
| International management in the conservation and sustainable use of oceans and marine resources for sustainable development. | [37,111] | Is there participation in international conventions for the management of the conservation and sustainable use of oceans and marine resources? |
| Participation in international agreements for the management of innovation in sustainability. | [24] | Is there participation in international agreements for the management of innovation in sustainability in organizations in the region? |
| National and international management in the conservation and sustainable use of oceans and marine resources for sustainable development. | [111] | Is there participation in national or international conventions for the management of the conservation and sustainable use of oceans and marine resources? |
| Promotion of sustainable guidelines in favor of society based on innovation. | [42,77,112,113] | Is there promotion of sustainable pro-society guidelines based on innovation in the region? |
| Establishment of support networks for the promotion of sustainable models in the region. | [35] | Are there communication networks between the different stakeholders involved in sustainability in the region? |

**Table 2.** *Cont.*

| Simplified Variable | Reference | Questions |
| --- | --- | --- |
| Search for balanced relations between the state, civil society, and organizations to achieve sustainable institutional development. | [79,110,114] | Are there mechanisms in place in the region to link the state, civil society, and non-profit organizations to achieve sustainable institutional development? |
| Higher education systems allow the flow of knowledge with the support of institutions that contribute to research, development, and dissemination. | [5,7,21,24,47,84,88–91,95] | Does the region have education systems that contribute to the development of attitudes and cognitive capacities that allow the acquisition of knowledge for decision making, industrial and economic development, based on the collective benefit and the environment? |
| Development of public policy tools that, through regulations, incentives, or mechanisms that motivate actions or behaviors of agents, contribute to environmental protection. | [48,83,85,115] | Does the region have regulatory systems or control mechanisms for environmental protection? |
| Improve the management of information on the status and pressures of forest resources, as a support for the development of actions aimed at the administration and sustainable management of the country's forests. | [13,81,83] | Does the region know of its forest resource, and do they have tools to protect it? |
| Development of innovative activities, technologies, and infrastructures that contribute to the production of new content. | [12,13,32,91] | Does the region have sufficient activities, technologies, and infrastructure to support the production of new content? |
| Development of innovations in the region. | [12,13] | Does the region have methods that contribute to the production of new developments for the sustainability of the region? |
| Management of the rational use of natural resources, the protection, and conservation of ecosystems, and the reduction of pollution, to protect the environment. | [13,37,79,83,95,109] | Does the region have sustainable guidelines in favor of environmental protection and conservation? |
| The string of value sustainable. | [78,110] | What percentage of the value chain of products and services developed in the region can be considered sustainable? |
| Undertakings in economic projects based on sustainability. | [7,31,78,89] | What percentage of ventures in the region are based on sustainability? |
| Reduction of cultural and linguistic differences in favor of sustainability. | [94] | Are there strategies to reduce the cultural and linguistic gap between the different stakeholders in the region? |
| Compliance with national regulatory standards for sustainability development. | [24,94] | Does the region comply with national regulatory standards for sustainability development? |
| Implements sustainability awareness strategies for the communities in the area of influence. | [47] | Are strategies implemented to sensitize the communities in the area of influence on sustainability in the region? |
| Construction of technological innovations to strengthen sustainability in industry or society. | [5,86,95,110,116,117] | Are there technological innovations for strengthening sustainability in the region's industries or society? |
| Promotes quality education, research, and dissemination of knowledge in innovation for sustainable purposes. | [5,89] | Are quality education, research, and knowledge dissemination in innovation for sustainable purposes promoted in the region's educational organizations? |

**Table 2.** *Cont.*

| Simplified Variable | Reference | Questions |
| --- | --- | --- |
| State of the region's economy. | [16,24,84,86] | Based on national standards, how is the economy in the region? |
| Innovation in sustainable regulations. | [5,47,85] | Is the region at the forefront of sustainable development regulations, and is it developing systems for their application and compliance in its processes? |
| Financing of scientific and innovation capacity for the development of new markets. | [81,85,91,108,118] | How is the financing of scientific and innovation capacity for the development of new markets? |
| Availability of resources or tax relief for the development of more sustainable production models. | [47,48,84,89,95] | What is the availability of tax relief in the region for the development of sustainable production models? |
| Framework conditions and favorable environment. | [5] | Is the region's economic environment conducive to sustainability? |
| Sustainable development planning with the environment, technology, and the different sectors contributing to the process, taking into account multiple planning factors such as training, exploration, dismantling of erroneous knowledge, and organizational development. | [79,92] | Does the region have planning systems in place that allow for a transition between current and sustainable processes? |
| Complementary capabilities and dynamics to the market. | [31] | Are the region's market dynamics oriented towards a sustainable consumer? |
| Tracking and quantification of resources for SDG compliance. | [84] | Are the region's resources invested in achieving the SDGs? |
| Application of technological innovation focused on the development of proposals for the implementation of renewable energies as a response to environmental and sustainability challenges. | [23,89,107,115,119, 120] | Is the development of proposals for the implementation of renewable energies observed in the region? |
| Mobility and accessibility. | [5,12,21,83,109] | The region has adequate mobility management and planning systems, as well as accessibility, allowing for the adequate development of local sectors. |
| Circularity strategies. | [121,122] | Regarding circularity strategies in your region: have you applied circularity strategies? |

*5.4. Case Study on the Measurement of a Sustainable Innovation System in the Study Region in Colombia*

In the present study, it is evident that for the public opinion of those surveyed in the region composed of Quindío and Risaralda, both people with competence coefficients considered as experts (K greater than or equal to 0.7) and non-experts (K less than 0.7), there are still many opportunities to improvement in the implementation of sustainability in the context of innovation, given that none of the responses presented in the tool, with a consensus percentage higher than 50%, were presented as the best possible option (among the options presented) in the context of sustainability.

There is also a significant opportunity for the region evaluated (Quindío and Risaralda) to deepen the sustainability variables identified with lower ratings according to their related questions, which are presented in Table 3.

Ermilova et al. [23] mentioned the fundamentals of open knowledge when making any application of concepts, clearly according to what was found in the present research. The first step towards the path of improvement is to homogenize concepts and accurate measurement methodologies that allow working on conclusive data.

**Table 3.** Questions according to the associated sustainability variable (own elaboration 2021).

| Sustainability Variables | Associated Questions | Authors Who Relate the Variable | Region | Competition Coefficient K |
|---|---|---|---|---|
| International management in the conservation and sustainable use of oceans and marine resources for sustainable development | Is there participation in national or international conventions for the management of the conservation and sustainable use of oceans and marine resources? | [37,111] | Quindío | K ≥ 0.7 and K < 0.7 |
| Sustainable value chain | What percentage of the value chain of products and services developed in the region can be considered sustainable? | [78,110] | Quindío | K ≥ 0.7 |
| Undertakings in sustainability-based economic projects | What percentage of ventures in the region are based on sustainability? | [7,31,78,89] | Quindío and Risaralda | K < 0.7 |
| Financing of scientific and innovation capacity for the development of new markets | How is the financing of scientific and innovation capacity for the development of new markets? | [81,85,91,108,118] | Quindío and Risaralda | K ≥ 0.7 and K < 0.7 |
| Research and development systems that allow process improvement and the creation of new patentable inventions | Does the region have economic resources earmarked for the development of academic research? | [5,21,24,47,89–91] | Quindío | K ≥ 0.7 |

## 6. Conclusions

From the literature review process, it is possible to highlight the existing opportunity for the objective valuation of sustainability in an innovation context, given that a representative number of authors coincide in exposing the difficulties in finding a methodology or procedure for the measurement of sustainability, which, thanks to social and commercial pressure, is currently focused on valuation methods for industries or brands that wish to showcase their good practices as leverage for their sales. However, methodologies or studies that seek to measure the sustainability of a specific region have rarely been presented.

From the sustainability variables in an innovation system extracted from the literature review, it was found that 43 were economic, 33 were social, 47 were environmental, and 93 were classified as other; that is, they cannot be contained in one of the three pillars of sustainability.

Most of the authors reviewed consider the environmental variable to measure sustainability in innovation systems, and sustainability is becoming increasingly complex in its composition; therefore, the classic dimensions (environmental, social, and economic) are insufficient at present to support all the dynamics that involve its valuation or measurement. There is consensus in the studies analyzed on the potential of sustainability in the context of innovation as a driving factor of social functions, such as the use of natural resources and the production of goods and services, which have a significant impact on the economic growth of industries, institutions, and the region in general.

Based on the evaluation made by the experts on the questions created from the analysis of the variables, it is determined that the tool responds, as a guide, to the monitoring of factors that influence the measurement of sustainability in the field of innovation. Parallel to this, through the application of the developed tool in the regions, it was found that there is a great lack of knowledge on the topic by the respondents, low accessibility to sources of information, and a high percentage of disarticulation among policies, processes, groups, and institutions. This leads to the conclusion that the region evaluated (Risaralda and Quindío) has a low level of sustainability. Therefore, it is recommended that all those concerned with the well-being of the region, to join efforts to develop a more sustainable future for their communities.

Within the limitations encountered in this study, it was found that the number of responses could have been greater, especially to ensure a balance in the participation of actors from universities, corporations, state, interface entities, society, and other actors specifically related to sustainability issues, circular economy and regenerative technologies, among others. Additionally, it is also important to obtain additional complementary responses through interviews with the region's stakeholders, so they can reinforce the gaps or problems and propose solutions to address them.

In relation to future research, it is recommended to conduct the study in other regions to generate a base of gaps and good practices that can serve as input for public policy in terms of dynamizing the innovation system. It could also become a source of ideas for projects that can be financed and co-developed by regional stakeholders. New research can be conducted based on the results of the project, and the generation of conceptual and mathematical models can better explain the role of the participants and the information and knowledge flows among them, as well as face-to-face meetings, workshops, focus groups, and other participation strategies, where additional answers can be obtained to the diagnostic questions and possible recommendations, good practices, or strategies to close the gaps in the analyzed region.

Finally, in order to achieve optimal development standards, it is concluded that it is necessary to make a transition from Innovation Systems (IS) to Sustainable Innovation Systems (SIS), where innovating in a sustainable manner offers the possibility of increasing the competitiveness of markets at the global level, complying with international requirements and regulations.

As discussed several times in this study, assessing innovation is difficult, due to its complex system, measuring the impact of the system on sustainability is a highly challenging task, contrary to what has been stated above, this tool aims to reduce subjectivity by objectively assessing sustainability.

**Author Contributions:** N.M.L.S.: substantial contributions to the conception of the work; the acquisition, analysis, and interpretation of data for the work. V.V.S.: substantial contributions to the conception of the work; the acquisition, analysis, and interpretation of data for the work. J.F.G.S.: substantial contributions to the conception of the work; the acquisition, analysis, and interpretation of data for the work. J.W.Z.S.: substantial contributions to the conception of the work and revising it critically for important intellectual content. G.L.O.M.: substantial contributions to the conception of the work and revising it critically for important intellectual content. J.L.S.R.: substantial contributions to the conception of the work and revising it critically for important intellectual content. All authors have read and agreed to the published version of the manuscript.

**Funding:** This research received no external funding.

**Institutional Review Board Statement:** Ethical review and approval was not necessary for this study due to having informed consent from participants and due to not involve sensitive information(sensitive information is under-stood as those that affect the privacy of the holder or whose improper use can generate dis-crimination e.g., data related to health, sexual life, or biometric data).

**Informed Consent Statement:** Informed consent was obtained from all subjects involved in the study.

**Data Availability Statement:** The data presented in this study (variables) are available within the document with their corresponding references. For more information related to the methodology, refer to the corresponding author.

**Conflicts of Interest:** The authors declare that the research was conducted in the absence of any commercial or financial relationships that could be construed as potential conflict of interest.

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
