# Peer review of "Innovation Systems and Sustainability. Development of a Methodology on Innovation Systems for the Measurement of Sustainability Indicators in Regions Based on a Colombian Case Study"

_sustainability, doi:10.3390/su142315955_

Round 1

Reviewer 1 Report

Dear Authors:

The paper presented "Innovation Systems and Sustainability. Development of a methodology based on innovation systems for the measurement of sustainability indicators in a region" is interesting, but can be improved, so some recommendations are made.

The abstract is long, and it is recommended that the authors shorten it with the most relevant information, since this will make it easier for readers to determine whether the article is of interest.

The selection of the sample, or experts, should be clarified. It should indicate how many in each of the roles indicated in step 5. It is understood that a discretionary non-probabilistic sampling has been applied, but this should be clarified in the paper.

On the other hand, it would be interesting to include data on the level of development of the areas analyzed, as this would allow a better evaluation of the results obtained. A rural area based on livestock and traditional agriculture is not the same as an industrial area. It is therefore recommended to include an economic and social assessment of the areas under analysis. 

It should be clarified whether the questions are open or closed, and include more tables of results.

A better analysis of the study regions and a specific section on limitations and future lines of research should be included in the conclusions.

Best regards:

Author Response

Dear Reviewer,

We are deeply grateful for the comments given on the manuscript sustainability-1982432 and we have made the necessary revisions in order to address your concerns.

We have substantially reduced the summary, and we have deepened its structure towards the main findings of the research, emphasized the paper's purpose of creating a methodology that would help the different regions of the globe, and highlighted the most significant conclusions on the relevance of the study of innovation systems in sustainability.

Information related to the methodology was further specified. The respective adjustment was made, where it was specified why the questions were addressed in a closed manner and how they were reached from the methodology.

Finally, a more exhaustive analysis of the regions was included in the results reported, taking into consideration the factors specific to the region, the distribution of the interested parties that were used as a sample in the development of the methodology is deepened in information, where the level of knowledge Kc and the economy, education and government. The conclusions were extended with limitations of the analysis and recommendations for future research in other areas of interest.

We hope the revised manuscript will better suit the Sustainability Journal but are happy to consider further revisions, and we thank you for your continued interest in our research.

Sincerely,

The authors

Author Response

Dear Reviewer,

We are deeply grateful for the comments given on the manuscript sustainability-1982432 and we have made the necessary revisions in order to address your concerns.

We have substantially reduced the summary, and we have deepened its structure towards the main findings of the research, emphasized the paper's purpose of creating a methodology that would help the different regions of the globe, and highlighted the most significant conclusions on the relevance of the study of innovation systems in sustainability.

About “Introduction has not clearly revealed what is the urgency of Innovation Systems and Sustainability so that methodological development is needed” thank you for this excellent observation. The introduction section could use more structure and clear guidance for the reader. For this reason, we have added to the manuscript aspects such as clarify the importance of developing this tool, and why is the importance to evaluate sustainability, which resides in the opportunity it presents for small regions that do not have a large budget for assessment to be able to have an approach to valuation with an easily replicable system, based on current sustainability standards with innovation systems in mind.

Also, the question “What about the measurement of sustainability indicators that have been done previously, what are the gaps in this research?” this information, which had previously been covered  in the discussions and conclusions, have been extended in the introduction by indicating that previously only tools for the assessment of sustainability in industries or companies had been found but very little reference had been made to the study within geographical regions. A more technical approach is presented in the discussions where it is mentioned which variables were previously found by different authors in the literature review and what were the specific gaps.

It has also been made clear within the document that the purpose of this paper was to identify sustainability variables within the framework of the innovation system concept, as well as to propose a methodology for diagnosing regions and identifying their gaps in a sustainability-oriented innovation system. Therefore, the tool is determined to be a guide to the measurement of sustainability in the context of innovation in any region. A mapping was carried out based on the review of the literature related to various methodologies in Innovation and Sustainability Systems that have been applied so far, which delve into multiple relevant factors within the research of sustainable innovation policies.

Finally, the conclusions were extended with limitations of the analysis and recommendations for future research in other areas of interest.

We hope the revised manuscript will better suit the Sustainability Journal, but are happy to consider further revisions, and we thank you for your continued interest in our research.

Sincerely,

The authors

Reviewer 3 Report

“Innovation Systems and Sustainability. Development of a methodology based on innovation systems for the measurement of sustainability indicators in a region” – The topic is not interesting. There is no innovation. The authors tried to discover something new from LR, however, the contribution is very poor. I suggest the authors to specify the topic clearly, be more concise and focused and specify the region clearly rather than just saying “in a region”. Good luck!

Author Response

Dear Reviewer,

We are deeply grateful for the comments given on the manuscript sustainability-1982432 and we have made the necessary revisions in order to address your concerns.

We found your comments extremely helpful and have revised them accordingly. Therefore, we have updated the title of the manuscript so that the future reader will have a better understanding of the purpose of the paper, "Innovation Systems and Sustainability. Development of a methodology based on innovation systems for the measurement of sustainability indicators in any region based on a Colombian Study Case".

 It has also been made clear within the document that the purpose of this paper was to identify sustainability variables within the framework of the innovation system concept, as well as to propose a methodology for diagnosing regions and identifying their gaps in a sustainability-oriented innovation system. Therefore, the tool is determined to be a guide to the measurement of sustainability in the context of innovation in any region.

We made special mention of the fact that the importance of developing this tool to evaluate sustainability, resides in the opportunity it presents for small regions that do not have a large budget for assessment to be able to have an approach to a valuation with an easily replicable system, based on current sustainability standards with innovation systems in mind.

Finally, additional information on the methodology was provided to confirm the rigorousness with which the study was conducted, as well as to explain why this region of Colombia was taken as a study case.

We hope the revised manuscript will better suit the Sustainability Journal but are happy to consider further revisions, and we thank you for your continued interest in our research.

Sincerely,

The authors.

Round 2

Reviewer 1 Report

Dear Authors

The work presents an improvement with the modifications made. Some aspects could be improved, but it is considered sufficient.

Best regards:

Author Response

Dear Reviewer,

We are deeply grateful for the comments given on the manuscript sustainability-1982432, We have done our best to bring the best version of this article to the journal.

In the process of continuous improvement, we made considerable professional updates to the wording and grammar to offer the reader the best possible experience; We hope you enjoy this new version and that it meets the journal's high standards.

Sincerely,

The authors

Reviewer 3 Report

The revised version is better. Still, you need minor revision in editing the key contribution of the research in the conclusion section. 

Author Response

Dear Reviewer,

We are deeply grateful for the comments given on the manuscript sustainability-1982432, We have done our best to bring the best version of this article to the journal.

In the process of continuous improvement, we made considerable professional updates to the wording and grammar to offer the reader the best possible experience; In the conclusions, the purpose of the article has been further discussed.

We hope you enjoy this new version and that it meets the journal's high standards.

Sincerely,

The authors
